# Weeds in Cover Crops: Context and Management Considerations

Barbara Baraibar [1,2,*], Charles M. White [1], Mitchell C. Hunter [3], Denise M. Finney [4], Mary E. Barbercheck [5], Jason P. Kaye [6], William S. Curran [1], Jess Bunchek [1] and David A. Mortensen [7]

[1] Department of Plant Science, Penn State University, University Park, PA 16801, USA; cmw29@psu.edu (C.M.W.); williamscurran@gmail.com (W.S.C.); jbunchek@hotmail.com (J.B.)

[2] Department of Horticulture, Fruit Production, Botany and Gardening, University of Lleida, 25003 Lleida, Spain

[3] American Farmland Trust, St. Paul, MN 55108, USA; mhunter@umn.edu

[4] Department of Biology, Ursinus College, Collegeville, PA 19426, USA; dfinney@ursinus.edu

[5] Department of Entomology, Penn State University, University Park, PA 16801, USA; meb34@psu.edu

[6] Department of Ecosystem Science and Management, Penn State University, University Park, PA 16801, USA; jpk12@psu.edu

[7] Department of Agriculture, Nutrition and Food Systems, University of New Hampshire, Durham, NH 03824, USA; David.Mortensen@unh.edu

\* Correspondence: barbara.baraibar@udl.cat

**Abstract:** Cover crops are increasingly being adopted to provide multiple ecosystem services such as improving soil health, managing nutrients, and decreasing soil erosion. It is not uncommon for weeds to emerge in and become a part of a cover crop plant community. Since the role of cover cropping is to supplement ecosystem service provisioning, we were interested in assessing the impacts of weeds on such provisioning. To our knowledge, no research has examined how weeds in cover crops may impact the provision of ecosystem services and disservices. Here, we review services and disservices associated with weeds in annual agroecosystems and present two case studies from the United States to illustrate how weeds growing in fall-planted cover crops can provide ground cover, decrease potential soil losses, and effectively manage nitrogen. We argue that in certain circumstances, weeds in cover crops can enhance ecosystem service provisioning. In other circumstances, such as in the case of herbicide-resistant weeds, cover crops should be managed to limit weed biomass and fecundity. Based on our case studies and review of the current literature, we conclude that the extent to which weeds should be allowed to grow in a cover crop is largely context-dependent.

**Keywords:** ecosystem services; soil erosion; cover; nutrient management; trade-offs

## 1. Introduction

As our understanding of ecological processes in agroecosystems improves, cropping systems are increasingly viewed as comprising interdependent component parts (i.e., soil, crops, weeds, water, etc.) that interact to shape ecosystem processes that can be beneficial, neutral, or detrimental to human interests [1]. When a component of the ecosystem contributes to beneficial processes, it conveys an ecosystem service [2]. Sometimes a component or process contributes to an undesirable outcome and thus contributes to an ecosystem disservice. Examples of ecosystem services can be found in abundance in the literature, while examples of disservices are not as common [3,4]. Some services and disservices are clearly distinguishable. Food provisioning from a crop is widely recognized as an ecosystem service, while an insect pest outbreak would be seen as a disservice. However, there are many cases where the role of a component of the ecosystem is ambiguous. Intraguild predation is an example of such ambiguity, where beneficial insects (e.g., predators) consume not only insect pests but also other predators [5].

The dual provisioning of services and disservices may also occur with weeds. The standard view of most scientific literature is that weeds are unwanted components of an

agroecosystem or, at least, they should be managed to limit populations within the growing crop [6]. Among other disservices, weeds can decrease crop yields, host insect pests or plant diseases, mechanically interfere with harvest machinery, and contaminate harvested grains or plant material, decreasing feed and forage quality [7,8]. However, weeds and their seeds can also support services, such as hosting beneficial insects [9,10], pollinators [11,12], and birds [13] or increasing plant diversity at the field and landscape level [14–16]. Some of these services will also contribute to sustaining yields by increasing predatory control of pests [9,17]. In light of this duality, the role of weeds in agroecosystems has been fundamentally re-evaluated over the last decade [15,16,18–22]. Weeds constitute a significant portion of plant biodiversity in agricultural landscapes, particularly in predominantly arable cropping landscapes [18,22,23], but the net effect of weedy plants on cropping system performance is likely highly context-dependent [21,23,24]. For example, weeds emerging simultaneously with an annual crop are much more likely to result in a yield reduction than those emerging weeks later [25]. Similarly, weed species identity, herbicide resistance status, or potential seed rain will play a role in determining the perniciousness of a weed community within a given system.

Weeds can also occur in cover crops, which are crops planted on otherwise fallow land to provide various ecosystem services [26,27]. Cover crops are typically not harvested as a cash crop, though some are harvested as forage or grazed by livestock. The benefits most often desired from cover crops include increased soil organic matter, optimized nutrient management, soil conservation, and weed suppression [28–30]. In some circumstances, weeds may be detrimental in cover crop stands, but, in other situations and unlike in most cash crops, weeds may also provide the same benefits desired from cover crops [20,31]. We are not advocating that weeds should be promoted within cover crops but instead, we aim to explore the potential benefits and harms that these species, which are common in cover crops (see Section 3), are contributing to the system. Despite the uncertainties regarding the benefits and disservices provided by weeds in cover crops, the subject has received little attention

In this paper, we first discuss the disservices and services associated with weeds in cover crops and subsequent cash crops, and present the current management strategies to manage weeds in cover crops. We then present two case studies and draw on published data to demonstrate how weeds can augment services provided by fall-seeded cover crops in annual grain rotations. Finally, we discuss the importance of context when making decisions about managing weeds in cover crops.

## 2. Disservices from Weeds in Cover Crops and Potential Services

Farmers may strive to minimize weeds in cover crops for a variety of reasons. Excluding weeds from cover crops ensures that weeds do not limit the provisioning of services by the cover crop species which were intentionally selected and planted. For example, nitrogen (N) provisioning may be limited if a legume cover crop is infested with non-legume weeds. Weed-infested cover crops can also contribute to the build-up of the weed seed bank if weeds produce seeds within the cover crop [26,32]. Although not consistent for all species, weed seed rain, in combination with soil seed bank densities, correlates with weed seedling recruitment and thus weed severity in the following crop [33,34]. Therefore, weed seed rain in a cover crop may pose a threat if those seeds can germinate and compete with cash crops at any other point in the crop rotation. This threat may be much larger if herbicide-resistant (HR) weeds are present in herbicide-based systems. Weeds within cover crops can also support insect pests or crop diseases that use weeds as a green bridge between cash crops and, therefore, their control may be viewed as a phytosanitary action [35,36]. For example, henbit (*Lamium amplexicaule* L.) and purple deadnettle (*L. purpureum* L.) can serve as hosts for soybean cyst nematode and are weed species that should be controlled in a cover crop if this pest species is a threat to the following cash crop [37].

Weeds can also influence the soil microbial community. Finney et al. [3] reported that fields left weedy during the winter were associated with a soil microbial community dominated by Gram-positive bacteria and actinomycetes, whereas microbial communities

in relatively weed-free cover crops like oats (*Avena sativa* L.) or cereal rye (*Secale cereale* L.) were dominated by Gram-negative bacteria, saprophytic fungi, and arbuscular mycorrhizal fungi (AMF). AMF can provide multiple ecological and agronomic benefits such as protection against root pathogens [38], increased tolerance to environmental stresses such as drought [39] and improved water uptake [40,41]. Similarly, Wortman et al. [42] reported a strong negative influence of early-season arable weed communities (primarily common lambsquarter (*Chenopodium album* L.), velvetleaf (*Abutilon theophrasti* Medik), redroot pigweed (*Amaranthus retroflexus* L.), field pennycress (*Thlaspi arvense* L.), and green foxtail (*Setaria viridis* (L.) P. Beauv.)) on AMF. Although AMF and other fungi and bacteria have been linked to enhanced soil health, our current knowledge does not allow us to define an "optimal composition" for a healthy microbial community or imply that the microorganisms related to weeds are providing a disservice to the system. What is clear is that the composition of the fallow period plant community can shape the soil microbiome and that the microbiome-mediating effects of such plant communities requires much more attention.

As primary producers, weeds also may be able to provide some of the same ecosystem services as those desired from cover crops. Even though the role of weeds within cash crops has been extensively examined and reviewed [15,18,43], little attention has been paid to the ecosystem service provisioning of weeds in cover crops. Most of the existing literature on this topic considers weeds as having an important role in providing food and habitats for wildlife and insects in agricultural landscapes [10,12,15]. Requier et al. [44] found that weeds growing in rapeseed (*Brassica napus* L.) and sunflower (*Helianthus annuus* L.) crops provided up to 40% of the honey bee diet between the mass flowering periods of those crops, thus playing a major role in pollinator conservation. Diehl et al. [9] also reported that compared with weed-free fields, weedy organic wheat (*Triticum aestivum* L.) fields supported a higher density and diversity of ground beetles (Coleoptera: *Carabidae*), many of which are important predators of insect pests and weed seeds. Increased populations of those predators can lead to reductions in yield losses due to insect pests [17].

When weedy plants are allowed to grow in the period of time between crops in an otherwise fallow field—that is, when they grow as cover crops—they may also provide beneficial services. In a meta-analysis, Wortman [20] estimated that weeds growing as cover crops in fallow fields could decrease the over-winter N leaching losses by as much as 60% compared with a bare fallow, thus serving a similar function to a N scavenging grass or brassica cover crop. Similarly, Wortman et al. [31] compared the impact of four spring-sown cover crop mixtures and a mixture composed entirely of weeds on soil moisture, nitrate availability, and yield of the subsequent crop and concluded that, despite lower biomass, the performance of the weedy mixture was comparable to cover crop mixtures and had a lower cost. Other studies have examined weedy species such as common couch (*Elytrigia repens* L.) and common chickweed (*Stellaria media* L.) as cover crops in maize (*Zea mays* L.) and soya beans (*Glycine max* (L.) Merr.), respectively, and concluded that they can decrease soil erosion and serve as an alternate management tactic compared with leaving the ground bare in these crops [45,46]. These benefits could be negated if the weedy cover crops contained noxious species that set seed, but in each of these cases, it was determined that the weed species present were not competitive within the crop rotation used. Certain weedy species with favourable phenology and agronomic qualities, such as field pennycress, are being bred for use as cover crops that can also be harvested for economic yield as an oil seed crop [47].

While these examples point to services arising from weedy species acting as cover crops, there has been no attempt to quantify the value of weeds growing in a cover crop, a condition much more likely to occur in farm fields. The focus of most weed-related cover crop studies has been the weed-suppressive effects of cover crops [26,48,49]. However, many cover crop stands contain substantial levels of weed biomass (see Section 3) Therefore, it is important to consider how weeds growing in a cover crop affect the provisioning of ecosystem services.

## 3. Current Status of Weed Infestations and their Management in Cover Crops

Weed biomass in cover crops differs among cover crop types, with grasses and mixtures that contain grasses being usually more weed-suppressive than brassicas and legumes [50,51]. Despite high levels of weed suppression, grasses, brassicas, and mixtures can harbour up to 500 kg ha$^{-1}$ of weed biomass in the fall, while weed biomass in legume cover crops can reach up to 1500 kg ha$^{-1}$ [26,50,51]. Increasing growing degree-days in the fall, can increase weed biomass in slow-growing cover crops because the increase allows the germination and growth of summer annual weed species. However, early cover crop sowing can also increase cover crop biomass in the fall and help to better suppress weeds [50,51]. Weed species composition in cover crops will vary depending on the time of sowing, cover crop identity, and biomass [26].

Currently, most farmers do not actively manage weeds in cover crops but there may be circumstances when they do. For example, farmers with herbicide-resistant horseweed (*Conyza canadensis* L. Conquist) are starting to chemically manage this weed in cover crop stands [52,53]. Similarly, some organic growers may mow, harrow, or terminate their cover crops earlier to prevent large weed seed production if weed abundance is high (Curran, personal observation). The choice of whether or not to manage weeds in cover crop stands is probably largely driven by each particular context, including the presence of hard-to-manage weed species, especially HR weeds, expected weed seed rain, and crop sequence. This will be discussed further in Sections 5 and 6.

## 4. Case Studies of Weeds in Cover Crops

We present two case studies to illustrate the ecosystem services that can be provided by weeds in cover crop stands. These case studies are based on two multi-year interdisciplinary field experiments conducted at a research station and on private organic farms in Pennsylvania, USA.

### 4.1. Weeds in Cover Crops Provide Ground Cover and Limit Erosion

4.1.1. Methods

This experiment was conducted within a multidisciplinary research experiment aimed at quantifying the ecosystem services and disservices provided by a range of cover crops in a wheat–maize–soya bean organic rotation at the Russell E. Larson Agricultural Research Center in Pennsylvania (USA) [27,54]. The "Cover Crop Cocktails" experiment (CCC) used a full entry design in which each phase of the rotation is present in every year. Ten cover crop monocultures and mixtures comprised the treatments, which were replicated in four blocks (Supplementary Table S1).

Within the CCC experiment, we measured percent ground cover in the 10 cover crop treatments (6 monocultures and 4 mixtures) and in a no-cover crop control. All plots were tilled before cover crops were sown. The control treatment received additional tillage to manage weeds during the fall and spring, except for an untilled subplot (3 × 3 m), which served as a measure of potential weed pressure in the absence of management and was where the measurements took place. The 10 cover crop treatments were planted in mid-August and replicated 4 times. Percent cover provided by cover crops and weeds was assessed visually in 3 0.25-m$^2$ quadrats per treatment in the fall (October and November) and spring. The contributions of cover crops and weeds to total percent cover were quantified separately in each quadrat. Measurements were repeated for 2 years in separate entries. Ground cover was used to model relative soil loss ($SL_r$) compared with a bare surface control [55]. The resulting $SL_r$ index ranged from 0 (100% cover, no soil losses) to 1 (bare ground, maximum soil losses) and was modelled as a negative exponential curve between vegetation cover (C, %) and erosion rates as follows:

$$SLr = e^{-bC}$$

where b is a constant which varies between 0.0235 and 0.0816, depending on the type of erosion, the vegetation, and the experimental conditions [56]. For this work, we held b constant at 0.0348, an average value for splash erosion [56]. We ran a linear mixed model using the "lme" procedure in R statistical software (R Development Core Team 2017) to compare $SL_r$, considering cover provided by the cover crop alone or in combination with weeds for each cover crop treatment. We used cover crop treatment and year as co-variables and block as a random factor.

### 4.1.2. Results and Discussion

When cover crops established slowly (in 2013, probably due to limited rainfall), mean fall cover crop ground cover was 35%; weeds provided an additional 22.5% cover. However, the cover provided by weeds differed across treatments, ranging from 11% in the cereal rye treatment to 40% in the medium red clover treatment (Figure 1). The most abundant weed species were common lambsquarters and chickweed [26]. Weed biomass reduced the modelled relative soil loss significantly ($p < 0.05$) in the canola, medium red clover (*Trifolium pratense* L.), and Austrian winter pea (*Pisum sativum* L.) monoculture treatments and in the fallow compared with cover crops alone in October (Figure 2A). In the fallow control, the cover provided by weeds reduced $SL_r$ by 55–61% compared to bare ground. By November, cover crops had accumulated more biomass compared with October and competed with weeds, thereby reducing the proportional contribution of weeds to total ground cover (Figure 1B). The mean soil cover provided by cover crops was 85.5%, and the potential for modelled relative soil losses was lower in November than in October for all treatments. However, weeds were still associated with a significant decrease in relative modelled soil losses in the canola and medium red clover treatments (Figure 2B).

In the spring of both years, weed biomass was lower in all treatments compared with fall, because summer annuals died over the winter and newly emerged weeds were just beginning to germinate. However, in the fallow treatment, the cover provided by weeds still decreased the modelled relative soil losses from 1 to 0.6.

Mechanical operations such as tillage and cover crop establishment can result in a pulse of weed seed germination [57]. If the timing of emergence and the resulting early weed seedling growth are faster than the cover crop, the niche intended to be filled by the cover crop is pre-empted by the rapidly growing weeds. This is especially true for slow-growing leguminous cover crops [26,54]. For instance, Austrian winter pea grows slowly when established in August in the Mid-Atlantic region of the US, leaving the soil vulnerable to erosion, but growth accelerates as fall progresses. A similar dynamic may occur with other slow-establishing legumes, such as hairy vetch (*Vicia villosa* Roth) or clover species [48,49]. In these cases, weeds may provide temporary cover that can prevent soil loss early in the season, while the contribution of cover crops to soil loss prevention may increase later in the season. Similarly, if cover crop establishment fails (e.g., due to low germination rates or non-permissive weather conditions), weeds could be critical for reducing soil erosion over winter. The combined average precipitation in October and November in the Mid-Atlantic region of the US can be substantial, approximately 75 to 100 mm (https://www.usclimatedata.com, accessed on 1 February 2021) and, therefore, soil cover by weeds during these months may prevent soil erosion before the cover crop becomes established.

This case study demonstrates that weeds can play an important role in reducing soil loss. Inter-annual variability in weed biomass and the rate of cover crop establishment and the resulting ground cover will influence the extent to which weeds can contribute to soil loss reduction. Future climate trends predict increases in precipitation in the fall that may lead to an increased need for soil cover during those months [58]. Weeds that germinate and establish more quickly than cover crops may be important contributors to soil protection in this changing climate scenario. Long-term data on the weed seed bank from the experiment at the end of the 3-year rotation (Baraibar, in preparation) will help

clarify whether the impacts of weed seed rain in cover crops hamper weed management in the long-term.

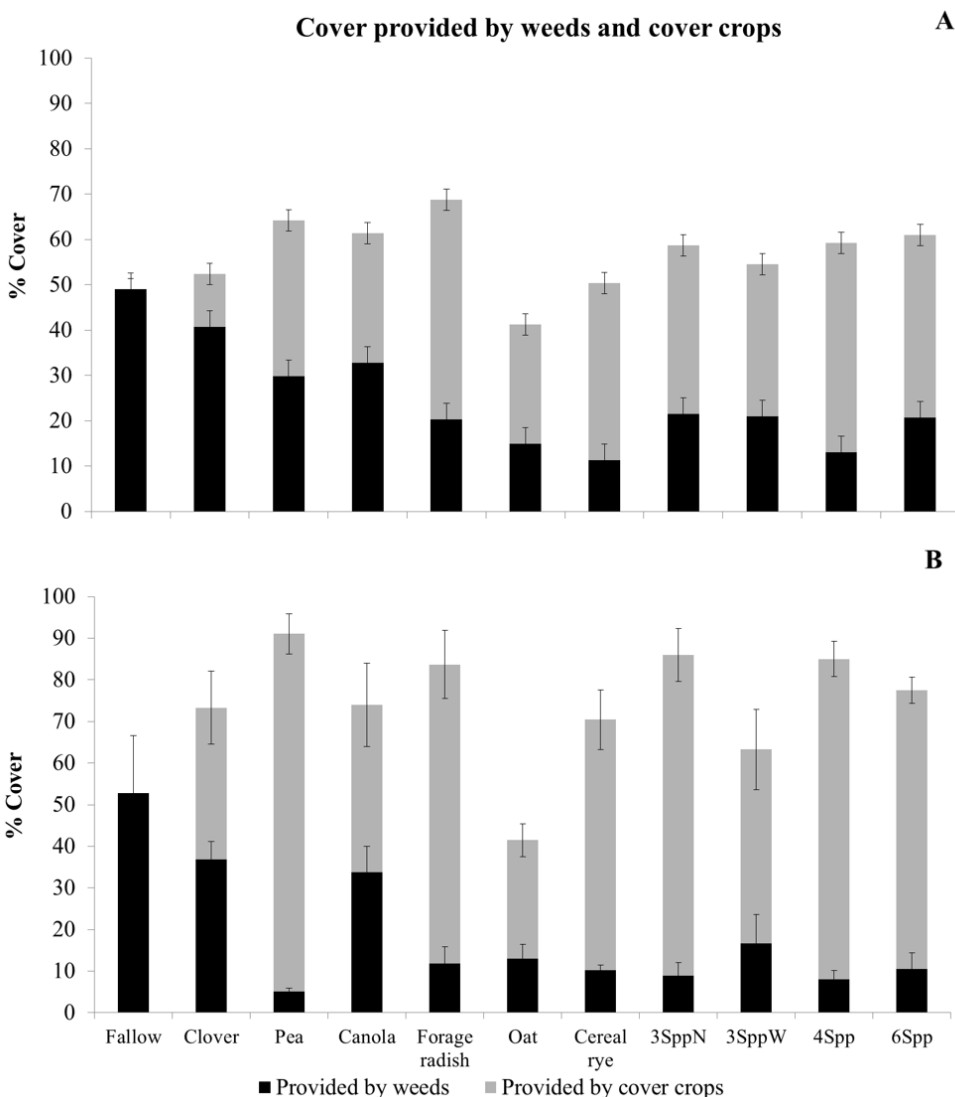

**Figure 1.** Ground cover provided by weeds growing in the cover crop (black bars) significantly increased total ground cover when compared with that provided by cover crops alone (grey bars) in October (**A**) and November (**B**) of 2013 in a field experiment located in central Pennsylvania, USA. Error lines correspond to ± standard error. No cover crop was seeded in the fallow treatment. 3SpN: cereal rye, medium red clover, Austrian winter pea; 3SppW: oats (*Avena sativa* L.), cereal rye, medium red clover; 4 Spp: cereal rye, canola, medium red clover, Austrian winter pea; 6Spp: cereal rye, oats, forage radish (*Raphanus raphanistrum* subsp. *sativus* (L.) Domin), canola, medium red clover, Austrian winter pea.

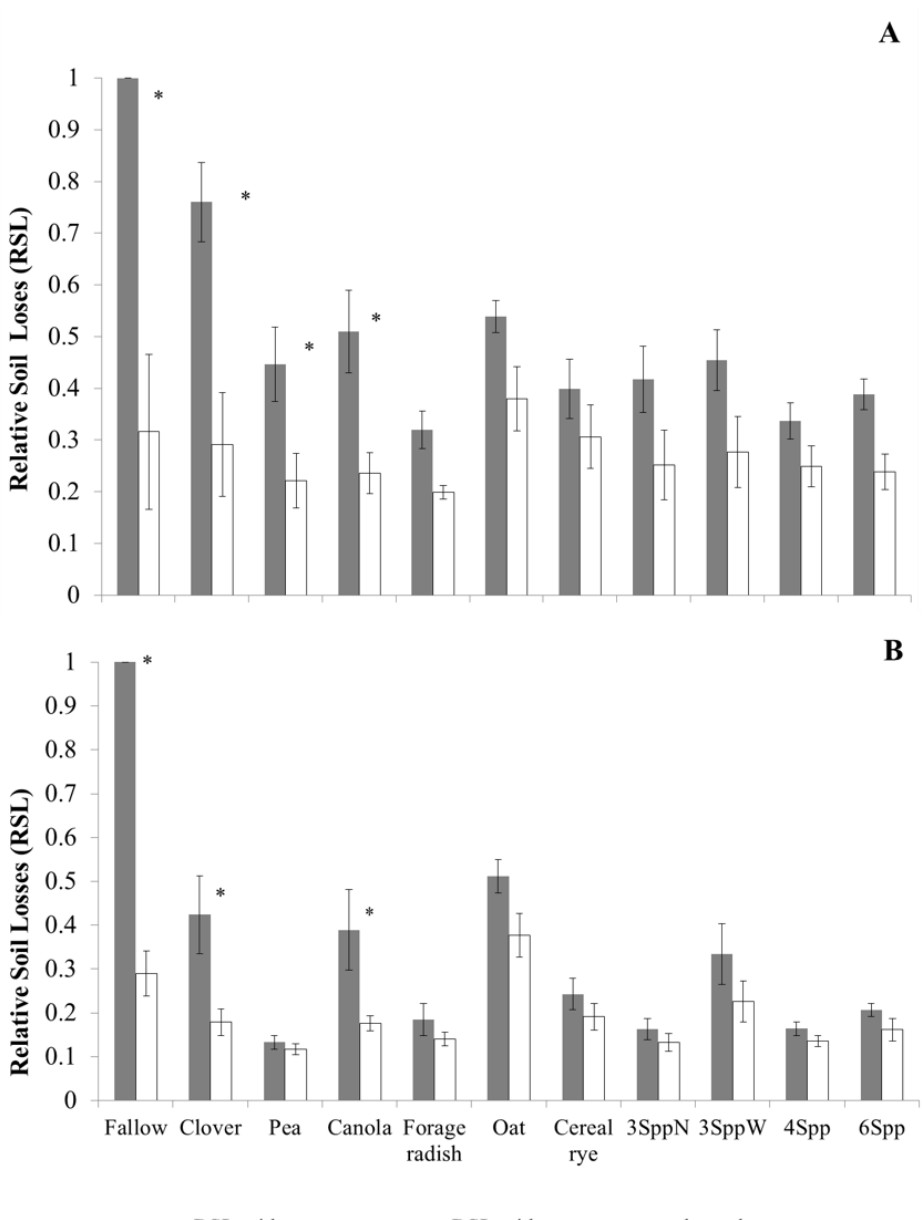

**Figure 2.** The additional cover provided by weeds significantly reduced the modelled relative soil losses (SL$_r$, soil loss under a specific vegetation cover compared with the soil loss on a bare surface) in some cover crop treatments (white bars) compared with when cover was provided by cover crops alone (grey bars) in October (**A**) and November (**B**) of 2013 in a field experiment located in central Pennsylvania. Error lines correspond to ± standard error. The relative soil losses index ranges from 0 (100% cover, no soil losses) to 1 (bare ground, maximum soil losses). No cover crop was seeded in the fallow treatment. 3SppN: cereal rye, medium red clover, Austrian winter pea; 3SppW: oats, cereal rye, medium red clover; 4 Spp: cereal rye, canola, medium red clover, Austrian winter pea; 6Spp: cereal rye, oats, forage radish, canola, medium red clover, Austrian winter pea. Asterisks (*) indicate significant differences at $p < 0.05$.

### 4.2. Weeds in Cover Crops Improve Nitrogen Management

4.2.1. Methods

The CCC experiment at the Russell E. Larson Agricultural Research Center in Pennsylvania was complemented by a parallel study at 3 organic grain farms across Pennsylvania [59]. Each farmer sowed a cover crop monoculture (a grass or a legume), a

3-species cover crop mixture of their own choice (Table 1), and a standard 4-species cover crop mixture (rye, *Secale cereale* L.; canola, red clover, and Austrian winter pea). The 4-species mixture treatment was also planted in the CCC experiment at the Russell E. Larson Research Center.

**Table 1.** Cover crop monocultures and mixtures seeded at three organic farms in Pennsylvania (USA) and at the Russell E. Larson Agricultural Research Centre (location 4)**.**

| Location | Monocultures | Three-Species Mixture |
|:---:|:---:|:---:|
| 1 | Rye | Crimson clover * + forage radish + Austrian winter pea |
| 2 | Red clover | Austrian winter pea + crimson clover + triticale |
| 3 | Ladino clover (frost-seeded) | Ladino clover + red clover + sweet clover |
| 4 | Rye | Austrian winter pea + red clover + rye |

* Crimson clover: *Trifolium incarnatum* L.; forage radish: *Raphanus sativus* L.; triticale: x Triticosecale Wittmack; Ladino white clover (*Trifolium repens* L. 'Ladino'); yellow blossom sweet clover (*Melilotus officinalis* L.).

At each location, we measured the aboveground biomass N content and C:N ratio of the cover crops and weeds in the fall (before the first killing frosts) and in spring (before cover crop termination) over 2 years [59]. Aboveground weed biomass N content was used as a proxy for weed N uptake, which directly affects the quantity of N available for leaching. Aboveground biomass was clipped from 2 $0.5 \times 0.5$-m quadrats randomly placed within each plot at the on-farm experiments and 3 $0.5 \times 0.5$-m quadrats at the research station experiment [59]. In each quadrat, weed biomass was separated from cover crop species biomass to determine dry weight and tissue C and N concentrations.

Potential nitrate leaching was measured from anion exchange bags buried at 30 cm during cover crop growth following the methodology of Finney et al. [60]. Nitrate leaching (in kg ha$^{-1}$) was regressed against fall and spring weed biomass N for the 2 seasons using the "lme" procedure in R statistical software (R Development Core Team 2017) to assess the role of weeds in preventing nitrate leaching.

4.2.2. Results and Discussion

In the fall, nitrate leaching under legume cover crops declined as N uptake by weed species growing in the cover crop increased (Figure 3). In one case, the tillage used to establish the seedbed for the three-species clover mixture stimulated a flush of weeds, mainly common chickweed, to germinate along with the clovers. In that weedy clover mixture, N leaching decreased to a level that was similar to that of a 4-species cover crop mixture comprised of two non-legume and two legume species. The weedy clover mixture also supplied N to the following maize crop at a similar level as the frost-seeded red clover monoculture, whereas N supply by the four-species mixture was lower because the non-legume cover crop species outcompeted the legumes [59].

Other studies have also acknowledged the role of weeds in altering the dynamics of nutrient retention and release. Finney et al. [60] grew eight cover crop monocultures and nine mixtures, and measured weed biomass and potentially leachable nitrate in the soil. When leguminous cover crops did not establish well and were invaded by weeds—for example, in fall-sown sunn hemp (*Crotalaria juncea* L.), soya bean, and red clover cover crops—weeds significantly decreased leachable N. In these cases, the existing weed seed bank provided "insurance" against suboptimal cover crop establishment and augmented the low N-retention capacity of the seeded legumes. More research is needed to determine the ability of specific weed species to scavenge N and contribute to the provisioning of this service.

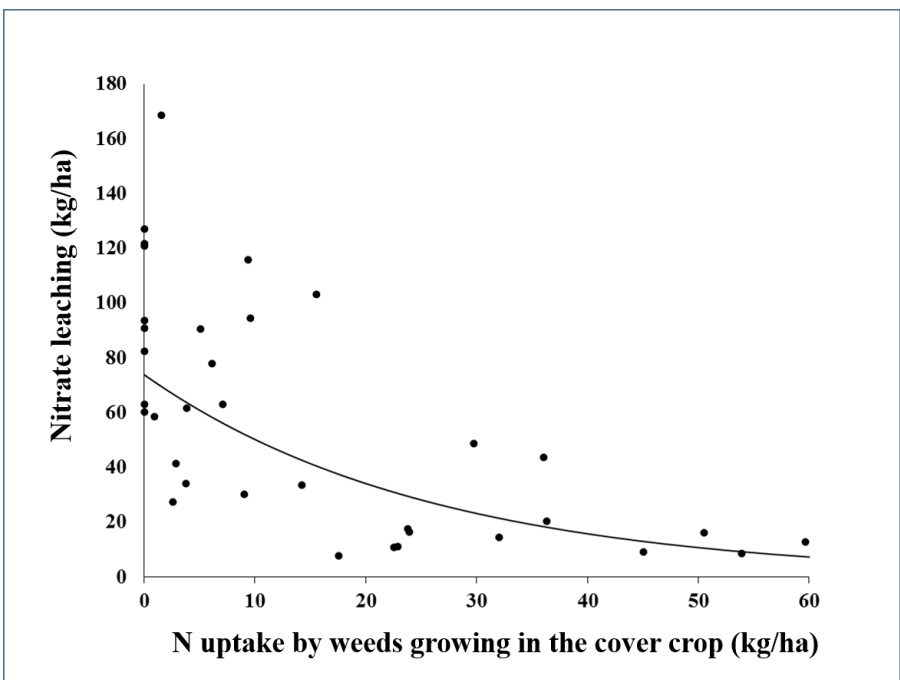

**Figure 3.** Nitrate leaching (kg ha$^{-1}$) in legume cover crops decreased as the N content of fall weed biomass increased (kg ha$^{-1}$). The equation for the regression is $y = 73.8\ e^{-0.0387x}$ and $r^2 = 0.54$ ($p < 0.001$). Data from White et al. [59].

Cover crops are widely used to prevent nitrate losses through leaching and to supply retained N to the following crop. While legume cover crops can supply from 50 to 370 kg N ha$^{-1}$ to a subsequent crop [61], they are less effective than non-legume cover crops at preventing nitrate leaching [62]. Planting cover crop mixtures containing legume and non-legume species is one strategy that could be used to manage the trade-off between N retention and supply [59]. One of the challenges with this strategy, however, is that the non-legume species in mixtures often outcompete the legumes [54], thereby raising the overall cover crop biomass C:N ratio and reducing N supply to the following crop [59,60]. This case study suggests that spontaneous weedy plants can, in some circumstances at least, complement leguminous cover crops without out-competing them. More research is needed to test under what circumstances and for which species these benefits may outweigh the possible disservices of weeds.

## 5. The Importance of Context

These case studies illustrate that weeds have the potential to enhance ecosystem services from cover crops. However, there are also many situations in which weeds may be undesirable in cover crop stands. Farmers must weigh the potential benefits and risks of weedy cover crops against the management costs and trade-offs of controlling weeds in cover crop stands, without sacrificing sustainable weed management. To date, there is no decision support tool that can help farmers in making these decisions, but choices should be largely be driven by the individual farm context. Salient aspects of this context include crop rotation, weed community composition (including HR weed species), weed management system, tillage system, and the relative N surplus or deficit (Table 2).

**Table 2.** Factors determining whether weeds growing in a cover crop may provide mainly services or disservices.

| | Weed Services Possibly Outweighing Disservices | Weed Disservices Possibly Outweighing Services |
|---|---|---|
| Crop rotation sequence | No cash crops in the rotation phenologically similar to cover crops | Phenologically similar crops in the rotation<br>Significant weed seed production<br>Low levels of weed seed predation<br>Weed is a potential host for pests of the subsequent crop |
| Weed identity or weed species traits | Less competitive weed species | Herbicide-resistant weeds<br>Very competitive weeds<br>Long-lived weed seeds<br>Perennial weeds |
| Cover crop use | As a green manure, the lower C:N ratio of weeds may improve N balance | As forage for livestock, unpalatable species should be avoided |
| Cover crop termination method | Chemical cover crop termination<br>Non-chemical cover crop termination (rolling) with high weed suppression potential in the following crop | Insufficient cover crop biomass to become an effective mulch against weeds in the following crop<br>Significant weed seed production |

Crop rotation will determine the impact that weed seed rain occurring in the cover crop phase of the rotation may have on weed problems in future cash crops. For example, winter annual weeds growing in a winter cover crop may pose a threat if a winter small-grain is present in the rotation, especially in organic systems [34]. Similarly, if summer annual weeds shed seeds within a late summer or fall-planted cover crop, subsequent summer weed control may be compromised. Because a portion of weed seeds may be consumed by seed predators or be subject to decay and fatal germination, not all weed seeds produced in a cover crop will necessarily contribute to future weed populations. Cover crops can increase populations of weed seed predators, which can decrease the number of seeds entering the weed seed bank [63,64]. Thus, weed seed production in cover crops may be at least partially offset by seed predation. However, more research is needed to elucidate the fate of weed seeds within rotations that include cover crops and to determine weed-specific economic thresholds for seed production within cover crop stands.

Weed species may also be important in determining the potential harm from weed seed rain in cover crops. Particular species may not be seen as problematic because of their low competitive ability in cash crops (e.g., chickweed, winter annual brassicas), whereas others may be particularly aggressive and should be more carefully managed. Marshall et al. [15] reviewed the potential services and disservices from 34 weed species and reported that, for example, a density of just two plants $m^{-2}$ of catchweed bedstraw (*Galium aparine* L.) could cause a 5% crop yield loss in a wheat crop, whereas 62 plants $m^{-2}$ of purple deadnettle were needed to cause the same loss. Similarly, perennial species such as nutsedge (*Cyperus escumlentus* L.), field bindweed (*Convolvulus arvensis* L.), or Canada thistle (*Cirsium arvense* L.) should not be allowed to thrive in cover crop stands, as their control may be difficult in later stages of the rotation, especially in organic farms. The same situation would apply in the case of HR weeds in conventional systems.

If populations of herbicide-resistant weeds are present in a cover crop that is part of a system using herbicidal weed control, they can be difficult to eliminate at termination and may eventually shed seeds that can contribute to the spread of resistant weeds. Herbicide-resistant horseweed is becoming increasingly problematic in winter grain crops such as wheat in the USA [65,66]. Similarly, herbicide-resistant summer annual weeds such as common ragweed, redroot pigweed, and Palmer amaranth (*Amaranthus palmeri* S. Watson) are increasing in summer annual crops [67,68]. Suppressive cover crops like cereal rye can help in managing these HR weeds by providing additional control when added to

herbicide programs [66,69]. In these cases, even if weeds may play a beneficial role within the cover crop, the context dictates that they should be effectively managed.

In winter cover crops, cover crop establishment and termination dates can affect the composition of the weed species pool that will co-occur with the cover crop. Management timing can be managed to a certain extent to avoid stimulating undesired species [70]. For example, delaying cover crop termination can reduce the weed density of early- and late-emerging summer annual weeds such as common lambsquarters, foxtail species, dandelion (*Taraxacum officinale* F.H. Wigg.), and common ragweed (*Ambrosia artemisiifolia* L.) [70]. Focusing on weed traits to determine the services and disservices that arise from weed communities can be a promising way to help determine which species may be problematic in certain cover crops [71]. Storkey [72] proposed a framework that used weed traits to identify weed species that combined a relatively low competitive ability with high importance for invertebrates and birds. The same framework could be used to determine which species could be allowed in cover crops and which should be actively managed.

Because cover crops are not generally harvested, yield loss within cover crops may not be an appropriate measure of the negative impact of weeds. However, cover crop establishment and quality may be compromised by weedy plants in some situations. Farmers using cover crops as forage for livestock may want to avoid certain weed species such as giant foxtail (*Setaria faberi* Herrm.), giant ragweed (*Ambrosia trifida* L.), and common cocklebur (*Xanthium pennsylvanicum* Wallr.), as they have been shown to be unpalatable [73–75]. Other weed species may be preferred by animals over forage species. For example, Bell et al. [76] reported that sheep foraged preferentially on weedy alfalfa compared with pure alfalfa stands, and that weeds did not affect the weight gain of lambs. For farmers incorporating cover crops into the soil as green manure, the impact of weed biomass on the overall cover crop C:N ratio is more relevant, as this measure of quality influences N supply to the subsequent cash crop. As illustrated by the case study, weeds can impact N dynamics and may have desired functional traits that support N management; however, more research is needed to quantify the effects of weedy species on the overall N supply and retention from cover crops.

## 6. Management Recommendations

In production contexts where low weed density and/or elimination of specific weed species in a cover crop are a priority, farmers can use several strategies to manage weeds within a cover crop stand. First, establishing a competitive stand by planting the cover crop at an appropriate time and at an adequate seeding rate will reduce weed establishment within the cover crop. Grasses are usually more weed suppressive than broadleaves, and mixtures that include grasses can obtain similar levels of weed-suppression than monocultures while also providing additional services [27,50,51]. Seeding rate and time of establishment also influence cover crop biomass production, which mediates weed biomass during the cover crop season [51,77]. Doubling or tripling the seeding rate resulted in greater weed suppression than seeding at a rate considered as standard [51,77]. Cover crop management also mediates the quantity of mulch present for weed suppression in no-till cropping systems [70,78]. Adequate mulch production can be particularly important in some cover crop-based organic rotational no-till systems that use a roller crimper to terminate the cover crop prior to no-till planting cash crops [78,79]. Second, the timing of cover crop termination, regardless of method, can also be leveraged to manage weeds within a cover crop [51]. When high weed seed production is a risk, for instance, terminating cover crops before viable seed production is reached can limit contributions to the weed seed bank. Third, control strategies such as tillage can also offset some of the results of an increased seed bank. Specifically, in high seed production scenarios, tillage after cover crop termination will place weed seeds at depths from which they cannot successfully emerge [80,81] and where they are exposed to soil microorganisms that cause seed decay [82,83]. If seed production by problematic weed species is not an issue, late termination in spring can reduce the germination of early summer annual weeds within

the cover crop because of an increased cover crop biomass [70]. Fourth, the use of tillage or herbicides to manage problematic weeds (e.g., perennial or resistant weeds) prior to cover crop establishment can also serve to prevent populations from thriving in a cover crop stand. Finally, in specific circumstances, chemical weed control may be indicated for weed control in cover crops, specifically if HR weeds that can cause severe problems in the following crops are present (e.g., herbicide-resistant horseweed).

## 7. Conclusions

The two case studies presented here revealed the strong beneficial effects of weedy plants found growing in a fallow period cover crop. These beneficial effects were particularly large in slower-growing cover crop species, where weedy plants added to the plant communities' ground cover and the resulting nitrogen retention and reduction in soil erosion. More research exploring the potential role of weeds as companion species to legume cover crops, elucidating the contribution of potential weed seed rain in the cover crop to future infestations, and understanding how weed species are filtered out by management practices will help leverage the potential services and disservices provided by weeds in cover crops and aid in decision-making.

**Supplementary Materials:** The following are available online at https://www.mdpi.com/2077-047 2/11/3/193/s1, Table S1: The composition and seeding rates (kg ha$^{-1}$) of the 11 cover crops used in the experiment (Section 4.1).

**Author Contributions:** Conceptualization: B.B., D.M.F., M.C.H., D.A.M., C.M.W., M.E.B., J.P.K., J.B. and W.S.C.; methodology: B.B., C.M.W., M.C.H. and D.M.F.; formal analysis, B.B. and C.M.W.; writing—original draft preparation, B.B.; writing—review and editing, D.M.F., M.C.H., D.A.M., C.M.W., M.E.B., J.P.K., J.B. and W.S.C. All authors have read and agreed to the published version of the manuscript.

**Funding:** This work was supported by the USDA National Institute of Food and Agriculture, Organic Research and Extension Initiative under Project PENW-2015-07433 (Grant No. 2015-51300-24156, Accession No. 1007156) and the National Science Foundation (Grant No. DGE1255832). Any opinions, findings, and conclusions or recommendations expressed in this material are those of the authors and do not necessarily reflect the views of the National Science Foundation.

**Institutional Review Board Statement:** Not applicable.

**Data Availability Statement:** All data are provided in the manuscript.

**Acknowledgments:** We are grateful to the entire Cover Crop Cocktails team from Penn State University and the staff of the Russell E. Larson Agricultural Research Centre for planting, managing, and assisting in data collection in our experimental plots, we would also like to thank the three reviewers who provided comments and suggestions to the manuscript.

**Conflicts of Interest:** The authors declare that they have no conflict of interest.

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
