# Peer review of "Weeds in Cover Crops: Context and Management Considerations"

_agriculture, doi:10.3390/agriculture11030193_

Round 1
Reviewer 1 Report
The manuscript by Baraibar et al. discusses an interesting and slightly controversial topic based on two case studies on 'cover crops and their interaction with weeds' from North America and relevant literature. The manuscript is well written but lacks details on certain aspects as highlighted in the attachment. The major concern I have is the fundamental argument that weeds may be beneficial in certain scenarios. This is true purely from an ecological point of view, but unfortunately, is often not the case when we talk about agro-ecosystems, especially crop production systems. Some of the arguments are well-supported from one-sided literature while others are mere assumptions. Having said that, I fully understand the context and its significance for future research in this domain. Therefore, authors are requested to consider the specific comments in the attached file. A separate section or 'current status of weed infestations and their management in cover crops' will be of great value. Feel free to accept or discard the suggestions but must provide feedback on each comment.

Author Response
Thank you for all your comments. We have addressed all your comments and provided more references where needed. Please see the attachment

Reviewer 2 Report
Abstract- this section looks well presented, and outlined clearly and scientifically.
- Add a comma in line 19: Here, we review services and disservices.........
Introduction-
- Add a comma in line 42: consume not only insect pests, but also other predators [5]
- Add a comma in line 54: particularly in predominately arable cropping landscapes [14,18,19], but the net effect of weedy plants......
- Cite this sentence (lines 43-46): The standard view of most scientific literature is that weeds are unwanted components of an agroecosystem, or at least, they should be managed to limit populations within the growing crop[ ].
- Cite this sentence(lines 58-59):Weeds can also occur in cover crops, which are crops planted on otherwise fallow land to provide various ecosystem services [ ].
- Cite this sentence (lines 62-64):In some circumstances, weeds
may be detrimental in cover crop stands, but, in other situations and unlike in most cash crops, weeds may also contribute to benefits desired from cover crops [ ]. - Clear objectives, good job (lines 67-71).
- Be consistent. Have second, third, etc. In line 73, you use the word "First" like this: First, excluding weeds from cover crops ensures that weeds do not limit the provisioning of......... Second, weeds.......Third, weeds.....
Methods- very clear, precise, and scientifically sound.
Results & Discussion- well presented and connected to the introduction/objectives and methodology.
Management recommendations- looks good, but be consistent when you start sentences like this (line 360): First and foremost, establishing a competitive stand.......
Second,.....
Third,......
Fourth or Finally,......etc.
Author Response
Thank you for your comments. Please see attachment for a point-by-point response

Reviewer 3 Report
The idea of this paper is interesting. I have a few comments.
ln 88-92. This observation is not necessarily a "disservice". Microbial communities are dynamic, it is evident that plants build up those microbes, which are most suitable for them. Changing plant species results in a shift in microbial species, without any negative feedback on crops. E.g. Hargreaves, S. K., Williams, R. J., & Hofmockel, K. S. (2015). Environmental filtering of microbial communities in agricultural soil shifts with crop growth. PLoS One, 10(7), e0134345. Please be clear here.
I am not convinced, how the authors calculated data in Figure 2. How were you able to calculate the relative soil losses for the cover crop only? How did you manage to receive a substantial number of repetitions for all the variants, taking into account different share of weeds per sampling area?
Chapter 3.2.2. is interesting, however, I would mention the results to the chickweed specifically, as the other weed species might not act similarly in catching the N leachates. That is strongly correlated with the root system depth, ability to uptake N, and pace of aboveground biomass development.
Please make the conclusion part more specific to your own findings.
Author Response
Thanks for all your comments. Please see the attachment for a point-by-point response

Round 2
Reviewer 1 Report
I am happy with the revised version of this manuscript.
Reviewer 3 Report
The authors significantly improved the manuscript.